# Streamlined structure determination by cryo-electron tomography and subtomogram averaging using TomoBEAR

Nikita Balyschew[1,2,9], Artsemi Yushkevich[3,4,9], Vasilii Mikirtumov[1,3], Ricardo M. Sanchez[1,2,8], Thiemo Sprink[5,6] & Mikhail Kudryashev[1,2,3,7] ✉

Structures of macromolecules in their native state provide unique unambiguous insights into their functions. Cryo-electron tomography combined with subtomogram averaging demonstrated the power to solve such structures in situ at resolutions in the range of 3 Angstrom for some macromolecules. In order to be applicable to the structural determination of the majority of macromolecules observable in cells in limited amounts, processing of tomographic data has to be performed in a high-throughput manner. Here we present TomoBEAR—a modular configurable workflow engine for streamlined processing of cryo-electron tomographic data for subtomogram averaging. TomoBEAR combines commonly used cryo-EM packages with reasonable presets to provide a transparent ("white box") approach for data management and processing. We demonstrate applications of TomoBEAR to two data sets of purified macromolecular targets, to an ion channel RyR1 in a membrane, and the tomograms of plasma FIB-milled lamellae and demonstrate the ability to produce high-resolution structures. TomoBEAR speeds up data processing, minimizes human interventions, and will help accelerate the adoption of in situ structural biology by cryo-ET. The source code and the documentation are freely available.

Cryo-electron tomography (cryo-ET) enables observation of natively preserved molecules in the context of intact cells[1]. The combination of cryo-ET with subtomogram averaging (StA) allows for obtaining angstrom-scale structures of macromolecules[2,3]. StA provided unique insights into the structure and function of viral and bacterial structural proteins[4,5], eukaryotic protein coats[6], actin filaments in sarcomeres[7] and even visualizing ribosomes inside intact bacterial cells[8] with bound small molecules[9]. Developments in hardware and software[10–15] allow obtaining higher resolution for the broader range of samples. In particular, recently several approaches suggested the refinement of non-

linear sample movement and of electron optical distortions allowing significant resolution improvement[9,16–18].

However, several major hurdles remain in making StA a mainstream method. First—the StA workflow includes multiple steps that are typically performed by specialized software packages[3] which requires special effort to interface[19]. Second—several steps in the StA workflow—tilt-series alignment and particle identification often require manual intervention. Third—optimal storing and processing of large amounts of 3D volumes requires large-scale computing infrastructure and is not straightforward even for expert users. Large amounts of intermediate

[1]Max Planck Institute of Biophysics, Frankfurt on Main, Germany. [2]Buchmann Institute for Molecular Life Sciences, Goethe University of Frankfurt on Main, Frankfurt, Germany. [3]In Situ Structural Biology, Max Delbrück Center for Molecular Medicine in the Helmholtz Association, Berlin, Germany. [4]Department of Physics, Humboldt University of Berlin, Berlin, Germany. [5]Core Facility for Cryo-Electron Microscopy, Charité-Universitätsmedizin Berlin, Berlin, Germany. [6]Cryo-EM Facility, Max Delbrück Center for Molecular Medicine in the Helmholtz Association, Berlin, Germany. [7]Institute of Medical Physics and Biophysics, Charité-Universitätsmedizin Berlin, Berlin, Germany. [8]Present address: EMBL Heidelberg, Heidelberg, Germany. [9]These authors contributed equally: Nikita Balyschew, Artsemi Yushkevich. ✉e-mail: mikhail.kudryashev@mdc-berlin.de

results such as tomograms, extracted particles, and metadata may occupy terabytes of hard drive space and need to be managed. Finally, most macromolecules occur in cells in limited copy numbers therefore in order to achieve meaningful resolution, it is necessary to record and process large amounts of tomograms. Excitingly, there is significant progress in speeding up tomographic data collection by recording tilt-series in parallel[20–22]. In order to process large amount of data, the processing software also needs to be designed for automation, yet with opportunities for user interventions where it is needed.

Here we present TomoBEAR (Basics of Electron Tomography and Automatic Reconstruction) which is an open-source workflow for mostly automated pipeline for structure determination from cryo-electron tomograms at scale. TomoBEAR interfaces commonly used software for cryo-ET and allow flexibility for users to develop pipelines for their molecules of interest. We demonstrate applications of Tomo-BEAR to four data sets reaching high resolution with minimal user input.

## Results
### Overall design and processing options
TomoBEAR is implemented as a modular pipeline runner, executing one module per one tilt-stack, a tomogram, or a set of particles.

TomoBEAR can be executed in parallel in computing environments such as multi-GPU workstations and/or high-performance computing clusters. TomoBEAR takes the output of data collection - movie frames stored on the hard drive as input, generates metadata and performs motion correction with MotionCor2[23], assembly of tilt-series, tilt-series alignment with IMOD[11], Dynamo[12] or AreTomo[24], defocus determination with GCTF[25] or CTFFIND4[26] and 2D correction of the contrast transfer function (CTF) followed by tomographic reconstruction using IMOD[27]. Up to this step, the workflow operates in a near-automated manner and could be used for live data processing during data collection (Fig. 1a, Supplementary Text 4). While several packages are incorporated into TomoBEAR, we intentionally attempted to minimize their number. As we do not redistribute the external packages with TomoBEAR and the users have to install them themselves, it is important to have fewer external packages due to the convenience of users and the ability to maintain the workflow.

For particle picking and subtomogram averaging several options are available depending on target molecules. If the molecules of interest can be identified in tomograms by template matching, it can be done automatically followed by particle extraction to the hard drive, generation, and execution of the Dynamo-based classification

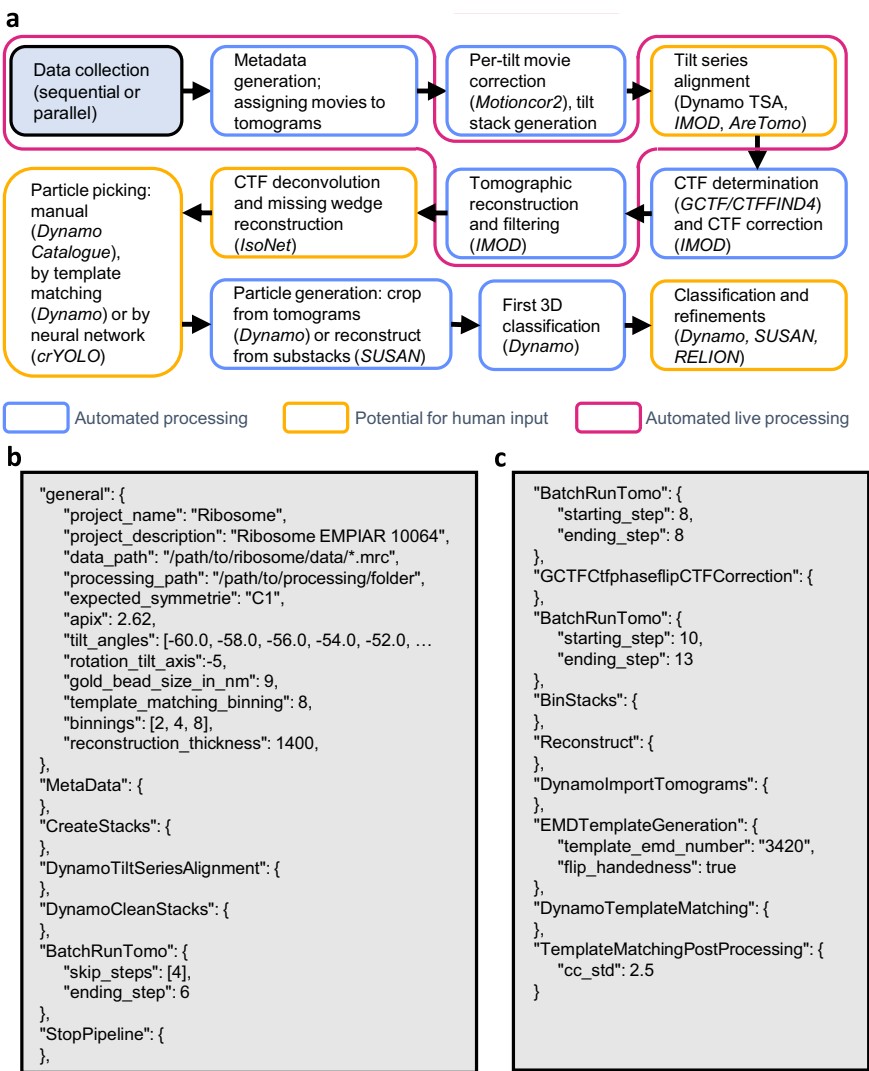

**Fig. 1 | A standard workflow for data processing in TomoBEAR. a** A flow diagram of data processing with TomoBEAR. Blue boxes outline the steps that are performed fully automatically, orange boxes may require human intervention. The steps highlighted in magenta represent the functionality of live data processing. **b**, **c** an example of a.json configuration file that was used for processing the EMPIAR-10064 data set (results below). Panel B contains the "general" section and the steps for processing up to the refinement of gold fiducials. **c** contains modules for CTF estimation and correction, tomographic reconstruction, and template matching.

project for initial subtomogram classification by multi-reference alignment using Dynamo[12]. Further alignment projects can be generated by selecting classes; upon changes in the desired binning levels - a selection of particles can be generated by direct reconstruction from tilt-series, without the need to reconstruct large unbinned tomograms.

When the molecules of interest cannot be picked by template matching−TomoBEAR creates data structures for the Dynamo Catalogue system[13]−a previously reported versatile tool for geometry-supported particle picking. For high-resolution structure determination metadata can be exported to RELION4[18] or the substack analysis tool SUSAN (https://github.com/KudryashevLab/SUSAN) based on projective cross-correlation[28,29].

TomoBEAR implements a "white box" approach that allows users to keep track of the used parameters and to monitor intermediate results. Modules can be re-run for the entire data set or for selected tilt-stacks/tomograms/particle sets providing an opportunity to test the pipeline on a subset of data to tune parameters and then process the entire data set.

## Automated tilt-series preprocessing and tomographic reconstruction

The user can define the execution parameters in a *.json* file (example in Fig. 1b, c Supplementary Text 3) which contains a *general* section describing the global parameters and tool-oriented sections with their specific parameters. A set of default parameters is provided in *defaults.json*, users can modify them by passing the parameters to the module description. Upon execution, the TomoBEAR output folder is populated by module-specific folders. Typically, each operation creates a sub-folder for each tilt-series/tomogram that can be inspected and upon a successful completion a SUCCESS file is created inside the corresponding folder. Upon a successful execution of the module for all the tomograms a global SUCCESS file is written into the module-specific folder. In order to re-execute the module, the global SUCCESS file can be deleted and the execution can be re-started.

The first operations of the cryo-ET and subtomogram averaging pipeline implemented in TomoBEAR are performed automatically. An output folder from a microscopy session (original movie frames) can be the input for TomoBEAR. Files are assigned to tilt-series (*SortFiles* module), drift is corrected (*MotionCor2* module) and the IMOD-compatible stacks are assembled (*CreateStacks* module). Batch alignment of tilt-series may be performed using gold fiducials with IMOD's BatchRunTomo[11] or Dynamo[12], or using patch tracking in IMOD or AreTomo[24]. The latter option is useful for aligning tilt-series from cryo FIB-milled lamellae. We found tilt-series alignment with the recently released tilt-series alignment algorithm DynamoTSA developed by Daniel Castano-Diez[12] to be robust, precise and well-performing computationally, therefore TomoBEAR uses it as default when gold fiducials are available. Importantly, DynamoTSA contains a robust estimation of whether the alignment was successful or not; this check allows the users to selectively inspect failed tilt-series. The resulting fiducial-based or fiducial-less alignments optionally can be inspected and potentially refined in a parallelly maintained IMOD project. The module *BatchRunTomo* allows the execution of the steps as they are defined in IMOD[11] (Supplementary Text 1). TomoBEAR uses the IMOD project for determining and optionally refining the final tomographic alignment parameters, CTF correction, and tomographic reconstruction. Initial defocus determination is performed automatically using GCTF[25] or CTFFIND4[26] and 2D CTF correction is performed by *Ctfphaseflip* from IMOD[30] in the *GCTFCtfphaseflipCTFCorrection* module. The fits of experimental and estimated power spectra can be examined in the processing folder. Aligned, binned, CTF-corrected tilt-stacks are produced and the tomograms are reconstructed at the user-defined binning levels. The tomograms can be post-processed by denoising with non-linear anisotropic diffusion[31] or by CTF deconvolution with IsoNet[32]

followed by a denoising and a filling of the missing wedge[32], this can be streamlined by TomoBEAR (Supplementary Fig. 1).

## Particle picking

Large proteins and lattices can be successfully picked by template matching[33]. In TomoBEAR we reimplemented the Dynamo[12] version of template matching to be executed on graphical processors, speeding up the processing 12−15 times, and developed auxiliary tools. A template for the search can be provided by a user as a file or it can be automatically produced from an EMDB entry by resampling to the voxel size of a tomogram and low-pass filtering it to conservative resolution. The cross-correlation map resulting from template matching can be post-processed, such as removing "large islands" and/or connected regions like membranes/edges (*TemplateMatchingPostProcessing* module). The user can inspect the cross-correlation (CC) maps and if the peaks are well-defined at the positions of the proteins of interest - particle extraction can be performed. For this, the positions of the top hits are extracted to a Dynamo-style table limited by threshold criteria (standard deviations compared to the mean CC value and/or maximal number of particles per tomogram; *TemplateMatchingPostProcessing* module). Dynamo-style table in this case contains initial orientations of particles that can be used as an input for local subtomogram alignment, classification, and averaging.

Another option for particle picking in TomoBEAR is crYOLO−a deep learning framework for particle coordinate prediction[34]. TomoBEAR interface includes crYOLO routines as configuration file preparation, input data pre-filtering, training, and prediction. After prediction, TomoBEAR allows extracting *Dynamo*-like particle tables per each tomogram for display purposes and for all tomograms together. Finally, when particles cannot be picked by template matching or the available neural network-based methods−a versatile set of particle-picking tools is available in the previously described Dynamo Catalogue system[13] which can be created with the *DynamoImportTomograms* module. Particle picking is generally performed in highly binned tomograms which can potentially be filtered[32,35] for visualization purposes.

As a result of particle picking/template matching the initial subtomograms are either extracted from existing tomograms or are reconstructed from tilt-series at the defined binning levels (*GeneratePartciles* module). For particles reconstructed from projections, 2D per-particle CTF correction of the tilt substack is performed taking the height of the particle in the tomogram into account. Particle reconstruction from projections stacks is performed on GPUs using the SUSAN engine (https://github.com/KudryashevLab/SUSAN)[36]. Additionally, to a significant speedup in the execution time, reconstruction of particles from projections eliminates the need to reconstruct large unbinned tomograms, saving data storage space.

## Subtomogram alignment, classification, and averaging

A multi-reference alignment and classification project with Dynamo can be generated and executed automatically (*DynamoAlignmentProject* module) with pre-calculated parameters. In our implementation, we use the particle picking templates as well as "noise traps" as references for the first multi-reference alignment project. This allows the separation of true positives/good particles and false positives/bad particles obtained by template matching, neural network-based particle picking, or semi-automated particle picking. Consecutive classification projects with Dynamo may be initiated after class selection by the user or several classification steps can be scheduled at the start of the processing (see example in Supplementary Text 3). The binning level can be reduced, in this case, a new set of particles will be produced either by cropping particles from reconstructed tomograms at lower binning or by direct reconstruction from tilt stacks ("SUSAN particles"). At each step of particle extraction, the particles are recentered. When a final particle set is produced, independent half-set

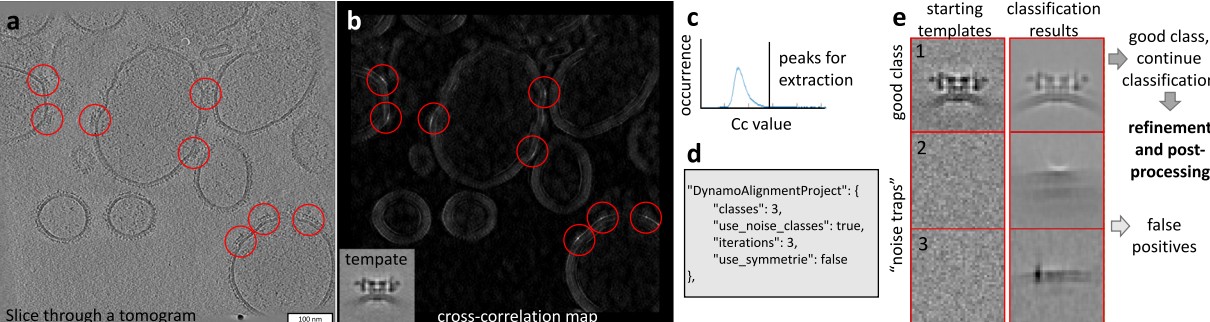

**Fig. 2 | Example of particle picking and subtomogram classification from a tomogram of an ion channel RyR1 imaged in native membranes (EMPIAR-10452). a** A slice through a tomogram with particles of RyR1 marked with red circles. **b** A result of the Dynamo-style cross-correlation search of a low-resolution template (shown in an inset) in a tomogram from **a** with the 15-degree angular search and C4 symmetry. **c** Selecting an offset for the CC values of the extracted peaks based on the histogram of the CC values in the map in **b**. **d** A TomoBEAR description of a classification project to run multi-reference alignment for 3 iterations with 3 classes, two of which are "noise traps". **e** Classification of particles extracted from the positions corresponding to the top values of the cross-correlation map starting with 3 classes, two of which are noise classes. The resulting first class contains particles that can be further processed, classes 2 and 3 - false positives.

refinement is performed in order to reliably assess the resolution. We call this processing workflow "conventional StA".

Generally, the automation of StA at the final steps of the refinement has limited utility as the user has a lot of flexibility to optimize particle sets, masks, filters, and other parameters. We tested the workflow for four data sets (below) and found that for simple objects with a well-tailored configuration, near-automated StA can reveal results close to optimal. Ultimately, Dynamo and similar approaches operating on 3D volumes that contain misalignments, have limitations in attainable resolution. Therefore, we implemented export to SUSAN (https://github.com/KudryashevLab/SUSAN) by creating the data structures and to RELION4[18] by creating *.star* files with tomograms and particle descriptions. The mentioned *.star* file can also be used for processing of hybrid StA data[37,38] with other single particle packages such as CryoSPARC[39].

## TomoBEAR utilities

Some operations of the cryo-ET/subtomogram averaging workflow require human intervention. As mentioned, TomoBEAR maintains a parallel IMOD project during tilt-series processing, which provides compatibility with the broad functionality of IMOD. This allows visual inspection of the processing steps, and, importantly, inspection and refinement of the gold fiducial-based or fiducial-less tilt-series alignment. A module *StopPipeline* can be added to the configuration file to stop automated execution for these purposes. We realized that often fiducial gold beads differ in sizes and specifying variable sizes helps automatic algorithms to succeed in tilt-series alignment. Therefore, multiple gold bead sizes may be input to the *general* module. In this case tilt-series alignment routine will try all the sizes consecutively until the first successful alignment.

Note that in the case of particle picking by template matching, false positive picks may be produced by gold beads or their shadows, edges of grid holes, edge artifacts from reconstructions, contaminations on ice surfaces, and other sources. TomoBEAR aims to minimize the amount of such false positives by erasing gold beads by default (IMOD implementation in 2D), smoothing image edges (as in Fig. 2a), and an option to erase grid edges from micrographs (*GridEdgeEraser* module).

TomoBEAR allows the generation of tomographic reconstructions "live" in order to probe the quality of the output data during data collection. For simplicity and speed-up TomoBEAR-live skips motion and CTF correction and performs reconstruction on binned input stacks. For initialization, users need to provide a *.json* file with the conventional set of parameters plus the expected number of images per tilt-series and the input data listening time threshold. We tested

live data processing to generate tomographic reconstructions of the previously reported HIV-1 GAG data set[4], EMPIAR 10164. For the tests, we simulated live data collection with a script that copied the original data to the simulated data collection folder updating the original timestamps. Time delays between virtually arriving frames and tilt series were set respectively to 5 and 30 s, imitating data collection. We measured times spent by TomoBEAR to process data from raw dose-fractionated movies up to reconstructed tomograms in both conventional (offline) and live modes (Supplementary Fig, 2). The simplified workflow aimed at the visualization of tomograms resulted in a three-fold speedup for processing data in live mode compared to the conventional one. The additional mini-guide on the "live" processing mode usage is available in Supplementary Text 4.

Data management and cleanup: as stand-alone packages, many single particle, tomography, and StA utilities produce temporary files that can be removed using the cleanup functionality of TomoBEAR during or after the execution.

## Benchmarking TomoBEAR

We benchmarked TomoBEAR on three previously reported and one original data set, the overview of input data characteristics, used tools, and results can be found in Table 1, and the corresponding per-tomogram processing time measurements are provided in Supplementary Fig. 2. First, we processed the tomograms of purified 80 S ribosomes imaged by cryo-ET (Khoshouei et al., 2017, EMPIAR 10064, mixedCTEM). This data set was already motion-corrected and tilt-series assembled. We performed automated conventional data processing with particle picking by template matching and subtomogram classification and averaging in Dynamo. The user inputs were: minor refinements of gold fiducial alignment and class selection during subtomogram classification with Dynamo. The resulting global resolution was 11.0 Å (Fig. 3a), while the resolution reported in the original publication was 11.2 Å. The processing time was ~1 h for preprocessing from assembled and motion-corrected stacks up to reconstruction and ~9 h for template matching and StA analysis.

Next, we recorded 60 tomograms of purified human apoferritin and performed data processing with TomoBEAR followed by export to RELION4 for StA. In this case, we performed tilt-series alignments by TomoBEAR without manual inspection or corrections. Particles were picked by template matching with reduced angular search range due to high symmetry of the target molecule. Particle set was imported to SUSAN and then to RELION4 after the initial alignment. Processing with RELION4 was done manually, two rounds of Refine3D, particle polishing and CTF refinement of 18k particles with O-symmetry in RELION4 resulted in a structure at a resolution of 2.8 Å, almost

**Table 1 | Summary for processing of the benchmarking data sets**

| DaZta set | Ribosome 80S (EMPIAR-10064) | RyR1 (EMPIAR-10452) | Apoferritin (EMPIAR-11543) | Ribosome 80S (EMPIAR-11306) |
|---|---|---|---|---|
| Input data dimensions | 3710 × 3710 × 59 | 3710 × 3838 × 41 | 5760 × 4092 × 31 | 4096 × 4096 × 35 |
| Nominal pixel size | 2.62 Å | 1.7 Å | 1.378 Å | 1.85 Å |
| Unbinned tomogram thickness, voxels | 1400 | 2000 | 1600 | 3000 |
| Number of tomograms | 4 | 52 | 60 | 179 |
| Manual interventions | Gold beads refinement, class selection | Gold beads refinement, class selection | Class selection | Alignment parameters optimization, class selection |
| Input data format | Assembled, motion-corrected MRC tilt stacks | Raw TIF movies | Raw TIF movies | Raw EER movies |
| Motion correction (global or patch-based[1]) parameters | Not needed | Local correction via patching: 5 5 15 | Local correction via patching: 7 5 15 | EER integration: upsampling: 2, fraction grouping: 34, exposure per fraction: 0.5; ft_bin: 2; patching: 7 5 15 |
| Alignment method | Fiducial-based (Dynamo + IMOD) | | | Fiducial-less (IMOD patch tracking) |
| CTF estimation/correction | GCTF/Ctfphaseflip (IMOD) | | | GCTF/Ctfphaseflip (IMOD), IsoNet deconv. (demo) |
| Reconstruction algorithm | weighted back-projection (IMOD) | | | |
| Particle picking method[2]: | GPU-enabled Dynamo TM | GPU-enabled Dynamo TM | GPU-enabled Dynamo TM | GPU-enabled Dynamo TM |
| parameters: | | | | |
| template | EMD-3420 | EMD-10840 | EMD-11603 | EMD-15636 |
| binning level | bin 8 | bin 8 | bin 8 | bin 8 |
| size of chunk | 463 × 463 × 175 | 479 × 463 × 500 | 512 × 720 × 200 | 512 × 512 × 375 |
| cone sampling/range | 15/360 deg | 10/360 deg | 9/90 deg | 10/360 deg |
| in-plane sampling/range | 15/360 deg | 10/360 deg | 9/45 deg | 10/360 deg |
| time per tomogram [3]: | 6m 2s | 15m 25 s | 0m 31s | 14m 53s |
| 3D classification | Dynamo MRA under TomoBEAR | Dynamo MRA under TomoBEAR[4] | RELION4 | RELION4 |
| | bin8 ~2h 26m | bin4 ~2h | bin4 ~3h | bin8 ~3h + 2h + 1.5h + 1.5h + 1h + 1.2h |
| Final refinement | Dynamo single-reference project under TomoBEAR bin4 - 04m 52s bin2 - 25m 47s bin1 - 5h 29m | RELION4 in bin2[5] ~2h RELION4 polished in bin1[6] ~11h | RELION4 in bin1 ~1d 19h 30m RELION4 polished bin1 ~4d 4h 20m | RELION4 bin8 ~3h Bin2 ~ 3h Bin1 ~ 20h Bin1 polished ~24h + ~12h + ~9.5h |
| Final number of particles and symmetry | 4003, no symmetry | 3169 · C4 symmetry | 20765 · O symmetry | 7575, no symmetry |
| Final resolution (at FSC=0.143) | 11.0 Å | 8.9 Å | 2.8 Å | 6.2 Å |
| Resources: | 16x CPU 2x GPU | 10x CPU 5x GPU | 16x CPU 2x GPU | 10x CPU 10x GPU |
| GPU hardware: | NVIDIA Tesla V100-PCIE-16Gb | NVIDIA Ampere A40-PCIE-48Gb | NVIDIA Tesla V100-PCIE-16Gb | NVIDIA Ampere A40-PCIE-48Gb |
| CPU hardware: | Intel(R) Xeon(R) Gold 6134 CPU @ 3.20GHz | Intel(R) Xeon(R) Gold 6248 CPU @ 2.50GHz | Intel(R) Xeon(R) Gold 6134 CPU @ 3.20GHz | Intel(R) Xeon(R) Gold 6248 CPU @ 2.50GHz |
| Environment: | Local execution in interactive cluster node | Local execution in interactive cluster node | Local execution in interactive cluster node | Local execution in interactive cluster node |

[1]Patching parameters are the number of patches along X/ Y and the neighboring patches overlap in percentage.
[2]TM is template matching.
[3]Given time is the average time for four tomograms.
[4]Different resources and hardware sets were used on the 3D classification step for the RyR1 (EMPIAR 10452) data set: Resources: 16x CPU + 2x GPU; GPU hardware: NVIDIA Tesla V100-SXM2-32Gb; CPU hardware: Intel(R) Xeon(R) CPU E5-2698 v4 @ 2.20 GHz.
[5]Different resources were used on the final refinement step for the RyR1 (EMPIAR 10452) data set: Resources: 40× CPU + 2× GPU; GPU hardware: NVIDIA Titan XP-PCIe-12Gb; CPU hardware: Intel(R) Xeon(R) CPU E5-2698 v4 @ 2.20 GHz; MPI processes: 9 × 4 threads per process.
[6]Different resources were used on the final refinement step for the RyR1 (EMPIAR 10452) data set: Resources: 40x CPU + 2× GPU; GPU hardware: NVIDIA Titan XP-PCIe-12Gb; CPU hardware: Intel(R) Xeon(R) CPU E5-2698 v4 @ 2.20 GHz; MPI processes: 5 × 2 threads per process.

reaching the Nyquist limit of 2.7 Å (Fig. 3b). We did not proceed with StA in RELION4 in super-resolution because of high requirements for computational resources (box size 400x400x400 voxels). For this data set processing with TomoBEAR up to export of the particle set to SUSAN was fully automated. The total runtime of TomoBEAR on 60 tomograms with the outlined hardware setup was around 1-1.5 days, the details are summarized in Table 1 and Supplementary Fig. 2. The time spent on subtomogram averaging consisted mostly of Refine3D runs, where the speed was limited by the available hardware and a relatively high number of particles needed to reach a resolution close to the Nyquist limit. In addition to that, the jobs spent some time queueing for the cluster resources. Nevertheless, the final map was obtained within two weeks from the day when the TomoBEAR processing was finished.

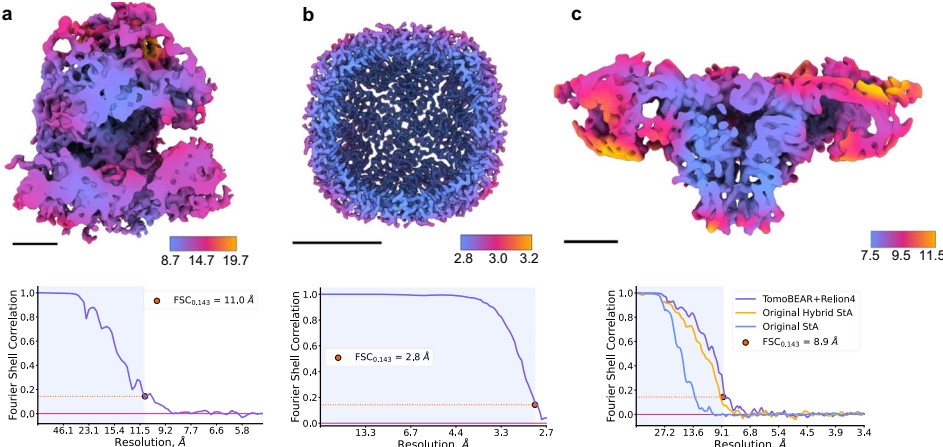

**Fig. 3 | Benchmarking performance of TomoBEAR. a** Processing of the tomographic data sets of purified ribosomes (EMPIAR 10064, mixed defocus): A structure at resolution 11.0 Å sliced in the middle of the reconstruction and colored according to local resolution. Black scale bars: 10 nm. Lower panels: detection of resolution based on Fourier Shell Correlation between independently refined halfsets. **b** Structure of purified human apoferritin imaged by cryo-ET in this study at the global resolution of 2.8 Å. The map is sliced in the middle of the reconstruction and colored according to local resolution. **c** Processing of tomograms of an ion channel RyR1 imaged in native membranes purified from rabbit muscle (EMPIAR-10452). The structure of RyR1 at a resolution of 8.9 Å sliced in the middle of the reconstruction and colored according to local resolution. Estimation of resolution based on Fourier Shell Correlation: the curves for the original processing of this data set from ref. 37 are in orange and blue, magenta - from TomoBEAR+RELION4.

Next, we re-processed our previously reported data set of the ion channel RyR1 (EMPIAR-10452) imaged in native membranes purified from rabbit muscle imaged by hybrid StA[37]. We performed automated tilt-series processing with minor refinement of gold fiducials followed by export to RELION4. Final refinement and particle polishing with RELION4 resulted in a structure at 8.9 Å resolution, slightly surpassing the originally reported resolution of 9.1 Å (Fig. 3c). Interestingly, the original structure was produced only from the untilted images recorded with a higher dose (15 e⁻/Å²). Here tilted images also contributed to the structure, slightly improving the resulting resolution, although the final particle set was also slightly different from the previously reported. While it is hard to accurately estimate the processing time spent for the structure in the original publication, it was in the order of months for particle picking, classifications, and refinements. In contrast, only a single round of Dynamo multireference alignment in the TomoBEAR environment was enough to obtain a homogeneous particle set suitable for downstream refinement. This allowed us to perform the subtomogram averaging and obtain the final map within one week, thus dramatically reducing the total processing time compared to the original mostly manual processing[37,40].

**Processing Data from FIB-milled Lamellae.** We next tested a workflow for processing cryo-ET data recorded on FIB-milled lamellae. As such samples typically do not contain gold fiducials, for such tomograms patch-tracking from IMOD[11] and alignment by AreTomo "local"[24] could be used. We benchmarked TomoBEAR on the recently reported data set of tomograms of HeLa cells milled by plasma-FIB at cryogenic conditions[41]. The data set EMPIAR-11306 contained 179 tilt series with an approximate thickness of 150-400 nm. We performed automated processing up to tomographic reconstruction using IMOD or AreTomo "local" for tilt-series alignment (Fig. 4a and Supplementary Fig. 1). Both methods resulted in tomograms of good visual quality.

We next performed ribosome identification for subtomogram averaging using two approaches: template matching and crYOLO[34] which has been used for particle picking in the original report[41]. Trained crYOLO was successful in identifying ribosomes, as validated by visual analysis (Supplementary Fig. 3). Template matching resulted in well-defined peaks (Supplementary Fig. 3) with lower-quality outputs in thicker tomograms. We extracted ~90,000 and ~180,000 top picks from crYOLO and template matching results respectively and

performed subtomogram classification in RELION4[18] outside of TomoBEAR. As the output of template matching contained initial orientations of the particles - we first performed classifications with local angular searches followed by a refinement, while for crYOLO particles we had to perform an exhaustive angular search. We performed the analysis for tomograms produced by IMOD patch tracking and AreTomo "local". The best structure at an overall resolution of 6.2 Å was obtained from template matching on tomograms aligned by patch tracking (Fig. 4c).

## Discussion

Here we report the workflow that enables at-scale processing of cryo-ET data for subtomogram averaging, which is becoming increasingly necessary. Most macromolecules are present in cells in limited copy numbers[42]; determining their structures in situ is only possible by processing large numbers of particles for alignment and averaging. While overexpression can be helpful in some cases, recording a high number of tilt-series is a generally applicable solution. Cryo-ET technology has become more accessible due to the larger microscope install base and the increasingly popular application of the "synchrotron model"[43]. Furthermore, the throughput of data collection dramatically increases by recording tomograms in parallel[20-22]. Therefore, recording large data sets becomes less critical than processing them. Manual processing requires time and expert knowledge of software and/or scripting and has risks of errors and inconsistencies. Suboptimal processing can not only limit the resulting resolution, but also unnecessarily amplify the amount of the used computational resources or hard drive storage. Therefore, the streamlining of processing with workflows like TomoBEAR will ease the entry barriers to the StA and will improve the quality of the resulting structures.

Several excellent workflows have been previously reported, among others: IMOD+BatchRunTomo[11], emClarity[16], tomoAuto[44], Dynamo[45], EMAN2[17], "M" combined with IMOD and RELION3[9], ScipionTomo[14]. The design of TomoBEAR is aimed at minimizing user intervention for large-scale data processing, focusing on high-resolution StA structures and maintaining flexibility. Another consideration was to provide the flexible functionality of cryo-ET data processing with a limited set of external tools to ease the maintenance requirements for the developers. Some important specific modifications have been implemented such as a good set of default values,

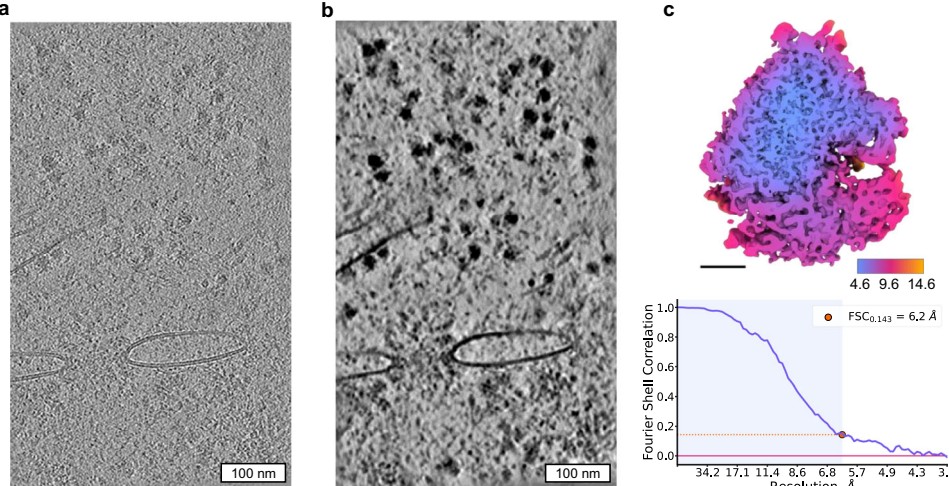

**Fig. 4 | Processing of FIB-milled lamellae with TomoBEAR. a, b** A tomographic reconstruction of the plasma-FIB milled HeLa cells data set containing 80 S ribosomes (Berger et al, 2023), EMPIAR-11306. The tomogram in **a** is reconstructed from a tilt stack aligned by patch tracking in IMOD **a**, and post-processed with IsoNet **b**. **c** The resulting 80 S ribosome structure at the resolution 6.2 Å sliced in the middle of the reconstruction and colored according to local resolution. Black scale bar: 10 nm. Lower panel: estimation of the resolution based on Fourier Shell Correlation between independently refined half-sets.

minimization of false-positives for template matching, etc. Indeed, we showed that with minimal user interventions, it is possible to process mid-scale data sets in a short time and to obtain up-to-date structures.

Here we presented four applications where minimal user intervention was applied and up-to-date structures were obtained for the benchmarking data sets of the conventional cryo-ET samples. For the new data set of purified human apoferritin, we could reach a resolution better than 3 Å, close to the sampling limit, suggesting that the used processing steps do not restrict the attainable resolution. A data set or RyR1 contains a membrane protein which is considered to be a more difficult target for particle picking and alignment to the average. We could process it with minimal user intervention to a resolution slightly higher than previously reported. For the 80 s ribosome from the cryo-plasma-FIB-milled samples, we obtained a resolution of 6.2 Å which was lower than the originally reported 4.9 Å[41]. The local resolution in our map was up to 4.6 Å, while the benchmarking dataset had a resolution of up to 3.8 Å. Since some tomograms were thick, not all particles had high CC scores after template matching, therefore we had to generously extract putative particles from tomograms (top 1000 hits per tomogram). This resulted in a large number of false positive peaks in the initial particle set and some good particles were lost during the multiple rounds of classification. As a result, the final particle stack had only ~7.5k particles compared to the ~17k in the original study which was probably a significant factor limiting the resolution. The data processing by Berger and colleagues utilized Warp-AreTomo-crYOLO-RELION3.1-M, while the essential steps in our workflow were performed by MotionCor2-IMOD-TM-RELION4.0 with the resulting structures at the comparable resolution. This case highlights the variability and flexibility of the workflows that can be employed for subtomogram averaging and the need for streamlining. Overall, the TomoBEAR workflow is applicable to a wide range of data sets and can routinely be used to process cryo-ET data. The post-processing approaches enable the refinement of processing imperfections at the final steps of structural determination[9,16–18]. The use of final refinement steps is therefore highly beneficial for the workflows as it allows more flexibility and a certain margin of error during the processing, making automated processing more robust.

As TomoBEAR is modular, the addition of further functionality is possible that can be used for optimizing the speed and performance of the StA modules (Supplementary Text 2). In particular, due to the high information content in cryo-ET data, currently, particle picking is a challenging step requiring significant computational time or manual work. Neuronal network-based particle pickers[46–48] and non-supervised tomogram annotation[49] tools could be further incorporated into the workflow pre-trained or with pre-defined parameters where possible. Furthermore, a streamlined workflow can be used to systematically evaluate the performance of algorithms for speed and the impact on the final structures. Importantly, in the future automatically processed subset of data could be used as training for neuronal networks that in turn can process large-scale data sets. In the future TomoBEAR would benefit from a closer integration with the Warp-RELION-M pipeline, currently, one such interface has been reported[19] or a web-based user interface similar to CryoSPARC[39]. Taken together, we believe that the use of TomoBEAR will lower the entry barriers into cryo-ET and sub-tomogram averaging and will speed up high-resolution structural analysis of macromolecules in their native state.

## Methods
TomoBEAR is a configurable and customizable software package specialized for cryo-electron tomography and subtomogram averaging written in MATLAB (Mathworks). TomoBEAR implements Basics for Electron tomography and Automated Reconstruction of cryo-electron tomography data. TomoBEAR is designed to operate on data in parallel where it is possible and to minimize user interventions and the need to learn the different software packages (MotionCor2, IMOD, GCTF, Dynamo) to be able to process cryo-electron tomography data. TomoBEAR contains a set of predefined defaults stored in the *defaults.json* file, which worked for most of the processing steps that we tested. However, if some parameters need to be modified for specific projects, the parameters can be further customized in an input.*json* configuration file. The description of the modules and the corresponding dependencies along with the tested versions are provided in Supplementary Text 1; TomoBEAR development design, notes for contributors, and module templates are available in Supplementary Text 2; a step-wise tutorial of the workflow for structural determination of the 80 S ribosome from benchmarking data set EMPIAR-10064 is provided in Supplementary Text 3; a mini-guide on the live data processing is available in the Supplementary Text 4.

### TomoBEAR as open-source software in research
The open-source status of the TomoBEAR project opens a possibility for the academic cryo-ET community to break the limits of a small

academic team by contributing to software testing, development, maintenance, and distribution. The broadness of the community experience and expertise helps to bring new ideas and overcome issues and initial design limitations, which prolongs software lifetime expectancy. In the end, the software is made first of all for user needs and the best way to deliver the requested functionality is to let users themselves participate in the development process, which is possible by open-source code distribution and open public communication channels in the TomoBEAR project. To support the community effort to contribute to the TomoBEAR, the software GitHub page contains contribution guidelines with a description of the necessary developer-level details. A short guideline on additional TomoBEAR modules development is also provided in Supplementary Text 2. The open-source TomoBEAR project as the software in research complies with the recently extended and adopted version of FAIR Principles[50] of scientific data management to the research software called FAIR4RS Principles[51]:

- TomoBEAR source code is fully deposited on the publicly accessible repository GitHub (https://github.com/KudryashevLab/TomoBEAR), is versioned and described with rich metadata (e.g., information on installation, usage and troubleshooting, citation, licensing, release and contribution notes);
- TomoBEAR is licensed under the open-source GNU General Public License version 3 and the beforementioned GitHub repository contains information about its development;
- The release version of TomoBEAR was deposited on the open research data repository Zenodo[52], it is accessible to both humans and machines;
- TomoBEAR operates and exchanges data with other software using standard cryo-ET input and output file types like TIF, MRC, XF, TBL, STAR, etc.
- TomoBEAR code includes cryo-ET domain-relevant dependencies to other software, which are listed for each module along with the tested software versions in Supplementary Text 1.

## Data processing details

**80S ribosomes (EMPIAR-10064, MixedCTEM).** This is a previously reported conventional transmission electron microscopy (CTEM) data set with mixed defocus values from the Volta phase plate report[53]. The raw tilt movies were already gain- and motion-corrected, and assembled into 4 tilt stacks which were used as input data for TomoBEAR. Further automated conventional data processing was performed for each step unless otherwise stated. Input tilt stacks were subjected to Dynamo tilt-series alignment routines to produce the fiducial alignment models that were imported to ETOMO from IMOD[27] and inspected manually, followed by the generation of the stacks. GCTF[25] was used to estimate defocus values for each tilt, and these values were input to *ctfphaseflip*[30] of ETOMO, followed by erasing of gold fiducials. CTF-corrected gold-erased aligned stacks were binned by a factor of 8, and bin8 tomograms were reconstructed and subjected to template matching, with the first template being a low-pass-filtered map of the 80 S ribosome from this data set (EMD-3420). 4490 top-hits were extracted and subjected to several subsequent multi-reference alignment projects in Dynamo each with 4 classes. Afterward, the best class containing 4005 particles was selected manually after visual inspection. Subsequent subtomogram classification and averaging were performed automatically using Dynamo within TomoBEAR. Particles for the final refinement were produced by direct reconstruction by SUSAN refinement for unbinned particles and resulted in 4003 particles averaged to produce the final map. The global resolution of the final map was estimated with the 0.143 criterion to be 11.0 Å, similar to the previously reported 11.2 Å with this data in the original publication[53].

**Purified human apoferritin (EMPIAR-11543).** Human apoferritin was produced according to the Leicester protocol[54]. The human apoferritin plasmid (LF2422) was freshly transformed into Escherichia coli BL21

(DE3) cells and protein expression was carried out in terrific broth supplemented with ampicillin (100 μg/ml). The bacterial cultures were grown at 37 °C and 80 rpm until OD600 reached 0.8, the heterologous protein expression induced by the addition of 300 μM isopropyl β-D-1-thiogalactopyranoside (IPTG) and incubated at 20 °C for another 15 h. The cells were harvested by centrifugation at 4000 g and 4 °C, collected, and frozen at −20 °C until needed. The cell pellet was resuspended in 50 ml lysis/wash buffer (50 mM Tris pH 7.5, 150 mM NaCl, 1 mM DTT) on ice before disruption using a microfluidizer (Microfluidics). To remove insoluble parts, the solution was centrifuged at 50,000 g for 45 min at 4 °C. The cleared supernatant was applied onto a prepacked Glutathione Sepharose 4B resin (Cytiva)-containing gravity flow column equilibrated in lysis buffer. To remove unspecifically bound proteins, the column was washed with 25 mL lysis/wash buffer. Bound protein was eluted using 30 mL elution buffer (50 mM Tris pH 7.5, 150 mM NaCl, 1 mM DTT, 20 mM GSH red). The GST-tag has been cleaved using recombinantly produced His6-tagged TEV protease during overnight dialysis at 4 °C against 3 L of lysis/wash buffer. A second Glutathione Sepharose 4B resin (Cytiva)-containing gravity flow column was used to seperate cleaved ApoF and GST. In order to remove TEV-protease and residual GST, size exclusion chromatography using a S200 column (Cytiva) and buffer SEC (50 mM Tris pH 7.5, 150 mM NaCl, 1 mM TCEP) was used. Fractions containing the pure protein have been concentrated, flash frozen in liquid nitrogen and stored at -80 °C until further use.

The final sample was diluted with a buffer containing 5 nm colloidal gold to a final concentration of 2.5 mg/ml. The sample was deposited on glow-discharged Quantifoil R1.2/1.3 Au grids, excess liquid was blotted by Whatman filter paper and the grids were flash-frozen in liquid ethane using a Vitrobot Mark IV (TFS) device. Grids were imaged on a TFS Titan Krios G3i operated at 300 kV equipped with a Gatan K3 detector and the Bioquantum energy filter aligned at zero-loss with a slit width of 20 eV. Tomograms were recorded as dose-fractionated movies using SerialEM 3.9[10] with the use of dose-symmetric tilt scheme[55] at a nominal magnification of 64.000 corresponding to a nominal counted pixel size of 1.378 Å. A total of 60 tomograms were recorded with an angular range of -45° to +45° with a 3° increment and a defocus range of -2 to -5 μm.

All the tomogram pre-processing steps were done under the TomoBEAR workflow unless otherwise stated. The processing was performed on a single node with 16 CPU cores and 2 GPUs in an automated fashion without user intervention. 1982 super-resolution non-gain-corrected tilt movies in TIFF format were sorted into separate folders by tomogram index and by tilt angle within these folders. Motion correction was done with MotionCor2 1.4.4[23] with 7 by 5 patches for local motion tracking, the last frame as a reference frame, and Fourier cropping of the output averages by a factor of 2. Raw stacks were generated from the motion-corrected images with the *newstack* program from IMOD[27]. Dynamo tilt-series alignment routines were used to generate the fiducial alignment models that were imported to *Etomo* from IMOD and the alignment models were refined, followed by generation of the aligned stacks. GCTF[25] was used to estimate defocus values per tilt, and these values were input to *ctfphaseflip*[30] from IMOD, followed by gold bead detection and erasion. CTF-corrected gold-erased aligned stacks were binned by a factor of 8, and bin8 tomograms were reconstructed and subjected to modified Dynamo template matching, with the first template being a bagel with dimensions similar to apoferritin. For this data set template matching was the last step completed with TomoBEAR, and subsequent subtomogram averaging was performed interactively. Cross-correlation peaks were extracted from the tomograms with a threshold of 7 standard deviations above the mean, comprising 21886 initial particle coordinates. Particle angles were randomized to get rid of the missing wedge, which resulted from the restricted angular search range in the template

matching step. The angular search range was restricted due to the high symmetry point group of the protein. After an initial alignment in SUSAN (https://github.com/KudryashevLab/SUSAN), particles were recentered and the *Dynamo* table was converted to a RELION4.star file with a TomoBEAR module. The project was imported to RELION4[18], the first round of 3D classification in bin4, and the selection of good classes led to 21068 final particles, which were subjected to 3D Refinement followed by CTF refinement and polishing. This final 3D refinement was performed with the box size 200x200x200 voxels[3] at counted pixel size. Pixel size was calibrated with an atomic model giving the final estimate of 1.331 Å/pix compared to the nominal value of 1.378 Å/pix. The global resolution was estimated with 0.143 criterion to be 2.8 Å. The final map was sharpened with a B-factor of -70 Å$^2$.

**Reprocessing the ion channel RyR1 data set (EMPIAR-10452).** This is a previously processed data set from the benchmarking of the hybrid StA data collection scheme[37]. The tomograms contain a 15 e-/Å$^2$ micrograph as a zero-tilt image. Tomogram pre-processing steps were done with the TomoBEAR workflow unless otherwise stated. All steps were run automatically without user intervention. 4099 non-gain-corrected tilt movies in MRC format were sorted into folders by tomogram index and by tilt angle within these folders. Motion correction was done with MotionCor2 1.4.4[23] with 5 by 5 patches for the high-dose zero-tilt image and with global motion tracking for tilted images. 100 raw stacks were generated from the motion-corrected images with the *newstack* program from the IMOD package[27]. Dynamo tilt-series alignment (https://wiki.dynamo.biozentrum.unibas.ch/w/index.php/Walkthrough_on_GUI_based_tilt_series_alignment) was used to generate and refine the fiducial models that were imported to ETOMO from IMOD and refined. At this point, we used the *StopPipeline* command, and the fiducial models were manually inspected in ETOMO and errors were corrected. Then TomoBEAR was restarted from the generation of the aligned stacks. GCTF[25] was used to estimate defocus values per tilt, and these values were input for CTF phase flipping with IMOD's *ctfphaseflip*[30], followed by gold bead detection and erasing. CTF-corrected gold-erased aligned stacks were binned by a factor of 8, bin8 tomograms were reconstructed and subjected to template matching with the modified *DynamoTemplateMatching* module, with the template being a resampled low-pass-filtered map of RyR1 from this data set (EMD-10840). Cross-correlation volumes were examined and only 52 out of 81 tomograms were kept. One hundred top-hits were extracted per tomogram and subjected to a multi-reference alignment project in Dynamo[12] with 3 classes. 3252 particles from the best class were selected and subsequent subtomogram averaging was performed interactively. After the initial alignment in SUSAN, particles were re-centered and the Dynamo table containing alignment parameters for particles was converted to a RELION4.*star* file with a dedicated TomoBEAR module. The EMPIAR-10452 data set consists of tilt-series collected with higher exposure on the untilted image, allowing for hybrid StA, however, our goal here was to benchmark tomogram preprocessing in TomoBEAR and therefore we decided to proceed with these data as conventional StA and keep all tilts. This way errors in tilt-series alignment or tomographic reconstruction can limit the resolution. The project was imported to RELION4[18], particles were subjected to 3D Refinement followed by CTF refinement and polishing, and then another round of 3D Refinement. Both 3D Refinements were done in bin1 with a box size of 320x320x320 voxels[3]. Importantly, upon import, the tomograms description STAR file column *rlnMicrographPreExposure* was modified to account for the uneven dose distribution between tilts. The global resolution was estimated with the 0.143 criterion to be 8.9 Å, similar to the previously obtained results with this data. Local resolution was estimated with RELION4.

**Processing of the plasma-FIB lamellae tomograms and the 80 S ribosomes (EMPIAR-11306).** Preprocessing of the EMPIAR-11306 was

done in the same way as for the previously described data sets, but with some exceptions. For motion correction with MotionCor2, the 343 or 336 EER frames were grouped by 34, so that 10 frames are rendered with 0.5 e-/Å$^2$ dose fraction. Tilt series alignment was done in IMOD with patch tracking using 500 × 500 size patches with 33% overlap along both axes. As for the previous data sets, GCTF[25] was used to estimate defocus values per tilt, and these values were used to perform CTF correction by *Ctfphaseflip*[30] from IMOD. Next, the CTF-corrected aligned stacks were binned by the factor of 8, and bin8 tomograms were reconstructed. Template matching with 360-degree search range and 20-degree step and the structure from the original report EMD-15636 was performed using the TomoBEAR. One thousand top hits were extracted per tomogram, giving 179,000 particles initially. These particles were directly imported into RELION4 with flipped defocus handedness and subjected to 3D classification into 8 classes with soft ribosome-shaped mask, initial low-pass filter 60 Å, fast sub-sets, circular mask with 370 Å diameter, solvent flattening, angular step 7.5 degrees and angular search within ±23 degrees. After each classification round, good and bad classes were separated and subjected to another classification round separately and then good classes from this sub-classification were merged. At each next classification round the angular search range was increased by 8 degrees. Five rounds of classification were done in total, giving a final count of 7575 particles. These particles were subjected to 3D refinement in bin8, bin2, and bin1, followed by Tomo Frame alignment and CTF refinement. Three consecutive rounds of Tomo Frame Alignment, CTF refinement, and Refine3D were performed. The final focused Refine3D was performed on a large ribosomal subunit, resulting in a 6 Å map according to the 0.143 threshold, which corresponds to a 6.2 Å resolution with a calibrated 1.9 Å pixel size value.

## Reporting summary
Further information on research design is available in the Nature Portfolio Reporting Summary linked to this article.

## Data availability
For the apoferritin data set, the raw microscope data was deposited in the Electron Microscopy Public Image Archive (EMPIAR) with the accession code EMPIAR-11543. The corresponding final subtomogram averages are available in the Electron Microscopy Data Bank (EMDB) under the deposition codes EMD-17232 [https://www.ebi.ac.uk/pdbe/entry/emdb/EMD-17232] (apoferritin), EMD-17272 [https://www.ebi.ac.uk/pdbe/entry/emdb/EMD-17272] (RyR1), EMD-18505 [https://www.ebi.ac.uk/pdbe/entry/emdb/EMD-18505] (80 S ribosome from the plasmaFIB dataset).

## Code availability
TomoBEAR is open-source, available on GitHub [https://github.com/KudryashevLab/TomoBEAR]; the 80 S ribosome data set and corresponding input preset files are available as a tutorial.

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

## Acknowledgements

The work is supported by the German Research Foundation (DFG, KU 3222/2-1), Sofja Kovalevskaja Award from Alexander von Humboldt Foundation to MK; the Heisenberg Award from the DFG (KU3222/3-1) to MK; support by the Josef Buchmann Family Foundation to NB, a starter fellowship from SFB807 (D.F.G.) to R.M.S. We thank the Core Facility for cryo-Electron Microscopy (CFcryoEM) of the Charité - Universitätsmedizin Berlin for support in the acquisition of the data. CFcryoEM was supported by the German Research Foundation (D.F.G.) through grant No. INST 335/588-1 FUGG and the Berlin University Alliance (BUA). We thank Daniel Castano-Diez, Kendra Leigh, Christoph Diebolder, Tobias Bock-Bierbaum, Vyacheslav Kralin, and Wolfgang Lugmayr for useful discussions, Uljana Kravchenko, Xiaofeng Chu, Giulia Glorani for testing the developmental versions and providing feedback, Özkan Yildiz and Juan Castillo from the Max Planck Institute for Biophysics for the IT support at the Max Planck for Biophysics and the high-performance computing team at the MDC for supporting our operation at the Max-Cluster.

## Author contributions

N.B. and A.Y. developed and tested the code, V.M. and M.K. contributed prototype code, V.M. processed the RyR1, apoferritin data sets, A.Y. and V.M. processed the purified and in situ ribosome data sets, A.Y. contributed to data analysis, T.S. produced the apoferritin data set, R.M.S. contributed to the TomoBEAR code. M.K. acquired funding, supervised the project, and wrote the manuscript with input from the other authors.

## Funding

## Competing interests

The authors declare no conflicts of interest.
