## [Peer Review File · Nature Communications]

Streamlined Structure Determination by Cryo-Electron Tomography and Subtomogram Averaging using TomoBEARReviewers' Comments:

Reviewer #1:

Remarks to the Author:

In this manuscript, Balyschew et al. present TomoBEAR – an open-source software package for cryo-ET and subtomogram averaging. Using cryoET datasets of ribosome 80S, apoferritin, and RyR1 in SR vesicles, the authors demonstrate that TomoBEAR streamlines structure determination by combining different software packages that are commonly used for cryo-ET and subtomogram averaging, including MotionCor2, IMOD, Gctf, and Dynamo. A strength of TomoBEAR is its customizability and modularity – new modules can be incorporated and modules can be easily applied to subsets of data in addition to full datasets. The authors have implemented the DynamoTemplateMatching module (template matching from Dynamo) on a GPU that is 12-15x faster than the CPU-based template matching implementation in Dynamo. The “white-box testing” approach that keeps track of parameters across multiple softwares is especially useful for users. The aim of TomoBEAR is to ease the entry barriers to subtomogram averaging and to improve the quality of structures by minimizing user intervention for large-scale data processing, with the option to conduct data processing along with data collection (TomoBEAR-live).

Major comments:

1. For the benchmarking experiments in this paper, the authors demonstrate using TomoBEAR MotionCor2, IMOD, Gctf, Dynamo, and SUSAN modules followed by export of a .star file into RELION4. The incorporation of AreTomo as a TomoBEAR module is mentioned in the text (but not used in benchmarking). The authors should demonstrate more varied processing pipelines in the benchmarking examples using additional widely used softwares in the cryo-ET community including convolutional neural network based applications for particle picking (e.g. crYOLO) and tomogram annotation (EMAN2).
2. The authors benchmarked TomoBEAR on ribosome 80S, apoferritin, and RyR1 in SR vesicles. The authors should incorporate an additional benchmarking dataset of in-situ cryo-FIB milled ribosomes to broaden the scope of this software to include data collected from lamellae (e.g. EMD-15807 Hoffman et al 2022 Nature Commun).
3. As mentioned in the discussion, WARP is a widely used software for cryo-electron tomography processing. The authors should discuss how and whether TomoBEAR can easily interface with WARP.
4. The main aim of this paper is to make structure determination from tomograms more straightforward. While the tutorials are excellent, it would be useful to have a supplementary video featuring a screen recording of the main steps of the TomoBEAR pipeline to demonstrate how user friendly the workflow is.
5. A web-based GUI for workflow visualization and analysis of intermediate processing steps with TomoBEAR would transform the field in a similar way to cryoSPARC for single particle cryo-EM. Can the authors discuss the feasibility of this for TomoBEAR?

Minor Comments:

1. Please include the Github link in the abstract for readers to quickly navigate to the TomoBEAR software for installation and use.
2. Line 29: In-situ should be omitted from the abstract if in-situ lamellae data is not included in benchmarking.
3. Line 69, 116, Fig1A pg 4, Table 1: Typo – change to MotionCor2
4. Line 87: Please include more information about SUSAN
5. Fig. 1A (pg. 4), Fig. 3 (pg. 9), Fig. S1: Please change the color scheme from red/green for colorblind readers.
6. Supplementary Text 2: Consider adding steps (e.g. Step 1: Download the dataset from EMPIAR) to make this tutorial easier to navigate.
8. Line 210: Change to “until”
9. Figure 3: Please add decimal points for axes labeling
10. Line 279: Typo – change to untilted

11. Line 319: Cite Wagner et al 2019
12. Table 1, 399: Typo – change to Gctf

Reviewer #2:

Remarks to the Author:

The manuscript by Balyschew et al presents TomoBEAR, a software package for streamlining the processing of cryo-electron tomography (cryo-ET) data. The package addresses an urgent need in the cryo-ET field for faster and more user-friendly processing tools, lowering the entry barrier for new users and increasing the efficiency of the technique. The manuscript presents a good overview of TomoBEAR, with its main features and design choices, as well as a couple usage examples. I specifically liked the usage of JSON files for handling the workflow metadata, the new GPU-enabled template matching feature, and the fact that it's an open source package. In summary, I'm happy to recommend the publication of this work in Nature Communications, pending only a few clarifications and minor corrections.

1 MAIN COMMENTS

As a potential TomoBEAR user and developer of cryo-ET data processing tools, I consider the manuscript and the package itself would benefit from addressing the following points:

1.1 CTFFIND4

Is there a reason why TomoBEAR does not support CTFFIND4 (<https://doi.org/10.1016/j.jsb.2015.08.008>) for defocus estimation currently? Since this free and open source package is one of the most widely used in the community, it would be nice to support it in the TomoBEAR pipeline. Another alternative to Gctf would be IMOD's ctfplotter, which might also be easy to include since IMOD is already well integrated into TomoBEAR.

1.2 Denoising

Denoising is an important part of modern cryo-ET pipelines, especially for processing in situ data. Does TomoBEAR integrates neural network denoising packages such as cryo-CARE (<https://doi.org/10.1109/ISBI.2019.8759519>) and IsoNet (<https://doi.org/10.1038/s41467-022-33957-8>)? If not, are there plans to include them?

1.3 Including new modules

Obviously the developers cannot support every software package available. So, considering the questions above, it would be good to explain if and how users can add new modules to the TomoBEAR pipeline. This is, in fact, more important than including support for the packages above mentioned.

1.4 Binning and reconstruction modules

From the supplementary material it was not clear whether TomoBEAR uses IMOD programs under the hood for the BinStack and Reconstruct modules, or if it has its own implementation of these tasks. I was able to check in the source code (and I praise the authors for sharing it), but please clarify in the modules description what is performed by TomoBEAR itself and what is interfaced from other packages.

1.5 Deposition of apoferritin dataset

As the authors used two other datasets deposited on EMPIAR for testing TomoBEAR, it would also make sense to deposit in that same repository the purified human apoferritin dataset they generated. This dataset could be useful for the testing and benchmarking of new tools by the community.

1.6 Cellular tomography data

The authors have tested TomoBEAR on 3 datasets, all of which were either from purified proteins or isolated membranes, containing gold beads for tilt series alignment. While it's clear that TomoBEAR

supports tilt series alignment without gold fiducials via patch tracking in IMOD or AreTomo, I would recommend if possible that the authors include one more dataset from in situ cryo-ET, i.e. from entire cells or FIB-milled lamellae, without gold fiducials. The authors may already have such a dataset in house, or could take one from the EMPIAR. This would strengthen the appeal of TomoBEAR to the in situ cryo-ET community even further, however, it is not essential for the acceptance of the manuscript. At a minimum, the authors should discuss what differences are expected when processing this kind of data. For example: are processing times significantly different? Can the automated workflow yield good quality reconstructions? How well is the new GPU-enabled template matching expected to work in crowded cellular environments? etc.

2 MINOR COMMENTS

2.1 SUSAN

Instead of referring to unpublished work, the authors could deposit the SUSAN source code, which is already available, in a citable repository such as Zenodo. I look forward to the manuscript describing the SUSAN package.

2.2 Reorder rows in Table 1

The rows "Number of tomograms", "Number of views per tomogram" and "Manual interventions" would be more useful if listed on the top of the table. Likewise, the hardware settings could be moved to the bottom.

2.3 Readability

The manuscript contains some redundant wording and sentences that seem incomplete, as well as a few typos. For better readability, I suggest the authors revise the text to minimize these issues. Below are some examples.

ABSTRACT

LINES 27-29

(...) to produce high resolution [structures?] with minimal human intervention. TomoBEAR is an open-source and extendable package [that might] accelerate the adoption of in situ structural biology by cryo-ET.

INTRODUCTION

LINES 41-42

allowed (...) allowing

RESULTS

FIGURE 1 LEGEND

LINE 107

highlighted in red highlight

LINE 96

module-specific sections with module-specific parameters

LINE 225

tilt-serie to tilt-serie

DISCUSSION

LINE 302

Remove "use of" (before "streamlining")

--

Reviewed by Ricardo D. Righetto

We thank the Reviewers and the Editors for the work on our manuscript. The revisions significantly improved the code and the manuscript. In our resubmission we provide tracked changes for the main text and the supplementary information together with the "clean" versions. Our point-by-point response is below.

Reviewer #1 (Remarks to the Author):

In this manuscript, Balyschew et al. present TomoBEAR – an open-source software package for cryo-ET and subtomogram averaging. Using cryoET datasets of ribosome 80S, apoferritin, and RyR1 in SR vesicles, the authors demonstrate that TomoBEAR streamlines structure determination by combining different software packages that are commonly used for cryo-ET and subtomogram averaging, including MotionCor2, IMOD, Gctf, and Dynamo. A strength of TomoBEAR is its customizability and modularity – new modules can be incorporated and modules can be easily applied to subsets of data in addition to full datasets. The authors have implemented the DynamoTemplateMatching module (template matching from Dynamo) on a GPU that is 12-15x faster than the CPU-based template matching implementation in Dynamo. The “white-box testing” approach that keeps track of parameters across multiple softwares is especially useful for users. The aim of TomoBEAR is to ease the entry barriers to subtomogram averaging and to improve the quality of structures by minimizing user intervention for large-scale data processing, with the option to conduct data processing along with data collection (TomoBEAR-live).

Major comments:

1. For the benchmarking experiments in this paper, the authors demonstrate using TomoBEAR MotionCor2, IMOD, Gctf, Dynamo, and SUSAN modules followed by export of a .star file into RELION4. The incorporation of AreTomo as a TomoBEAR module is mentioned in the text (but not used in benchmarking). The authors should demonstrate more varied processing pipelines in the benchmarking examples using additional widely used softwares in the cryo-ET community including convolutional neural network based applications for particle picking (e.g. crYOLO) and tomogram annotation (EMAN2).

Thank you very much for pointing out the advantages of the approach and the constructive comments. As a part of the response to this point and also to the comments of the second reviewer, we expanded the use of additional applications. We included Ctffind4 for CTF determination, IsoNet for deconvolution/filtering and filling of the missing wedge, crYOLO for NN-assisted particle picking and the functionality for FIB-milled in situ data. Interestingly, we contributed to the IsoNet codebase to overcome one of its important limitations - the original

code contained a constant angular range of the missing wedge of $-60...+60$. We added documentation and display items for Aretomo/Isonet/crYOLO in the Figures S2 and S3.

In general, we do not provide the external software in our distribution and the users have to install it themselves. Therefore, in order to make the workflow more robust we tend to minimize the number of external packages used in our workflows. This makes it easier to install for the users and the package ends up to be more robust with less sensitivity to the versions of the external software. Unsurprisingly, different packages and their different versions require different versions of CUDA, Anaconda, etc and an ambition to unify too many packages will cause execution errors and it will be impossible to maintain the software. Some hardware does not support certain versions of CUDA. Therefore, we rather focus on the functionality of the processing steps. In this respect, EMAN2 is an excellent full-workflow package, but we replaced many of its functions with the other packages or TomoBEAR code. We added a note about our design considerations to the text (end of the first paragraph of the results section)

2. The authors benchmarked TomoBEAR on ribosome 80S, apoferritin, and RyR1 in SR vesicles. The authors should incorporate an additional benchmarking dataset of in-situ cryo-FIB milled ribosomes to broaden the scope of this software to include data collected from lamellae (e.g. EMD-15807 Hoffman et al 2022 Nature Commun).

Thank you for the suggestion, we added the workflow for data processing for FIB-lamellae data focusing on the deposition of plasma-FIB tomograms of HeLa cells from Berger et al, Nature Communications, 2023 (EMPIAR-11306). This dataset contained a large number of tomograms and there was a ribosome structure available from it (EMD-15636). We introduced a workflow for processing of tomograms using patch tracking in IMOD (by batchruntomo) or in AreTomo for tilt series alignment, which was previously used for tomograms from cryo-FIB-milled samples. The workflow turned out to be faster and more automated than with the use of gold beads; the users could also attempt to use it for the conventional tomography samples.

We used this workflow for processing EMPIAR-11306 testing IMOD/AreTomo patch tracking for tilt series alignment followed by template matching / crYOLO for particle identification. We show that the tomograms can be reconstructed automatically with good quality (new Figure S2), the selected or all tomograms may be processed by IsoNet to perform CTF deconvolution to increase the low-resolution contrast and further to fill the missing wedge. For subtomogram averaging we got the best results using the simplest tools: patch tracking in IMOD + template matching + local classification and refinement in Relion4. The final resolution of 6.2 \AA with the local resolution up to 4.6 \AA was lower than in the

benchmarking dataset. We attribute it to the use of "Warp-Relion-M" workflow of the original paper; however we believe that a 6.2 Å structure serves as a proof of principle for the functionality of the workflow in its application to the cryoFIB/pFIB data. We wrote a section about processing cryo plasmaFIB data and commented on the usage of TomoBEAR.

3. As mentioned in the discussion, WARP is a widely used software for cryo-electron tomography processing. The authors should discuss how and whether TomoBEAR can easily interface with WARP.

Thank you, indeed our benchmarking on the plasmaFIB tomography showed that the original structure had higher resolution, although we had ~50% of the particles in our structure. We attributed it to better fitting software (Warp/Relion/M) in the Berger et al. It is possible that the Warp/Relion/M pipeline is more robust in refining tilt series alignment than the approach that we utilized. We added the discussion about it as well and pointed out that it would be useful to closer integrate tomobear with Warp/M. At the moment we are unable to integrate the Warp/M into the tomobear routines as the users would need a Windows machine for it.

4. The main aim of this paper is to make structure determination from tomograms more straightforward. While the tutorials are excellent, it would be useful to have a supplementary video featuring a screen recording of the main steps of the TomoBEAR pipeline to demonstrate how user friendly the workflow is.

Thank you for the valuable suggestion, we recorded a series of four short (8-12 mins) video-tutorials covering the TomoBEAR installation and setup, project configuration, execution and checking intermediate results as well as troubleshooting tips. We uploaded them on youtube (<https://www.youtube.com/@KudryashevGroup/playlists>), added the links to the GitHub page.

5. A web-based GUI for workflow visualization and analysis of intermediate processing steps with TomoBEAR would transform the field in a similar way to cryoSPARC for single particle cryo-EM. Can the authors discuss the feasibility of this for TomoBEAR?

TomoBEAR workflow would definitely benefit from a web-based GUI which could help to further decrease barriers between users and softwares. For example, this feature could let users easily access, visually inspect and interact with data during processing, allow smoother switches between (semi)manual and automated steps, monitor workflow execution progress, and etc.

We see the technical opportunity to develop an interactive web-app with a possibility to be hosted locally on workstations or HPC computing nodes to be able to use local computing resources and securely access sensitive research project data. However, it is very hard to fund using the academic funding mechanisms. We added a note at the end of the discussion session citing cryoSparc.

Minor Comments:

1. Please include the Github link in the abstract for readers to quickly navigate to the TomoBEAR software for installation and use.

Thank you, we updated the abstract

2. Line 29: In-situ should be omitted from the abstract if in-situ lamellae data is not included in benchmarking.

Thank you, we added the lamellae data and updated the abstract

3. Line 69, 116, Fig1A pg 4, Table 1: Typo – change to MotionCor2

Thank you, we updated it.

4. Line 87: Please include more information about SUSAN

We are working on releasing the package, to have a snapshot of a functional version we uploaded SUSAN on Zenodo: <https://zenodo.org/record/7950904>. In the current manuscript we do not use the high-resolution refinement with it and only SUSAN for pre-alignment and the reconstruction of subtomograms by skipping the reconstruction of full tomograms.

5. Fig. 1A (pg. 4), Fig. 3 (pg. 9), Fig. S1: Please change the color scheme from red/green for colorblind readers.

Thank you for an excellent comment, we updated the color palettes for Figure 3 and the new Figure 4.

6. Supplementary Text 2: Consider adding steps (e.g. Step 1: Download the dataset from EMPIAR) to make this tutorial easier to navigate.

Thank you, we updated it.

8. Line 210: Change to “until”

Thank you, we updated it.

9. Figure 3: Please add decimal points for axes labeling

Thank you, we updated it.

10. Line 279: Typo – change to until

Thank you, we updated it.

11. Line 319: Cite Wagner et al 2019

Now we use crYOLO as the method in the manuscript. The paper is cited and this sentence is about the potential further developments

12. Table 1, 399: Typo – change to Gctf

Thank you, we updated it.

Reviewer #2 (Remarks to the Author):

The manuscript by Balyschew et al presents TomoBEAR, a software package for streamlining the processing of cryo-electron tomography (cryo-ET) data. The package addresses an urgent need in the cryo-ET field for faster and more user-friendly processing tools, lowering the entry barrier for new users and increasing the efficiency of the technique. The manuscript presents a good overview of TomoBEAR, with its main features and design choices, as well as a couple usage examples. I specifically liked the usage of JSON files for handling the workflow metadata, the new GPU-enabled template matching feature, and the fact that it’s an open source package. In summary, I’m happy to recommend the publication of this work in Nature Communications, pending only a few clarifications and minor corrections.

Thank you very much for pointing out the utility of the manuscript and for the constructive comments.

1 MAIN COMMENTS

As a potential TomoBEAR user and developer of cryo-ET data processing tools, I consider the manuscript and the package itself would benefit from addressing the following points:

1.1 CTFFIND4

Is there a reason why TomoBEAR does not support CTFFIND4 (<https://doi.org/10.1016/j.jsb.2015.08.008>) for defocus estimation currently? Since this free and open source package is one of the most widely used in the community, it would be nice to support it in the TomoBEAR pipeline. Another alternative to Gctf would be IMOD's ctffplotter, which might also be easy to include since IMOD is already well integrated into TomoBEAR.

Thank you very much, we integrated CTFFIND4 as an option for CTF determination ("GCTFCtfphaseflipCTFCorrection" module).

1.2 Denoising

Denoising is an important part of modern cryo-ET pipelines, especially for processing in situ data. Does TomoBEAR integrates neural network denoising packages such as cryo-CARE (<https://doi.org/10.1109/ISBI.2019.8759519>) and IsoNet (<https://doi.org/10.1038/s41467-022-33957-8>)? If not, are there plans to include them?

Thank you for pointing out important functionality. We added two filtering options:

1. We implemented IsoNet for two possible uses: first is CTF deconvolution that amplifies the low frequencies providing additional contrast (new figure S2). The second use of the IsoNet is denoising and filling of the missing wedge which together can now be done via TomoBEAR. As no user input is needed for it - processing with IsoNet can be done automatically for the selected tomograms. We added a module IsoNet and described in the Supplementary Text 1 and on the tomoBEAR wiki.
2. We also added a simple filter non-linear anisotropic diffusion (NAD) as implemented in IMOD. It is now described in the section "Processing FIB-milled Lamellae for Structural Cell Biology" of the main text (as well shown in the new figure S2).

A TomoBEAR module was implemented to integrate the IsoNet functionality, including preprocessing (STAR file preparation, mask creation, CTF-deconvolution), training (refinement) and prediction. Users can design and execute sub-pipelines using those operations in order to produce several sets of data. The latest enables the possibility to conveniently produce train and test sets of data or conduct parameters optimisation which is essential for neural network applications.

CryoCARE has very difficult logistics and it would require a major refactoring of the code. We attempted to prototype it but because of the lower dose that two resulting half-tilt-series have, we could see that the alignment of tilt series becomes less reliable. The combination of IsoNet's CTF

deconvolution and filling of the missing wedge with the NAD filtering seems to produce nice filtering without the need to install an extra piece of software (Figure S2).

1.3 Including new modules

Obviously the developers cannot support every software package available. So, considering the questions above, it would be good to explain if and how users can add new modules to the TomoBEAR pipeline. This is, in fact, more important than including support for the packages above mentioned.

Thank you for the suggestion. We support the opportunity of the community to re-use the tomoBEAR code or to contribute new modules. We added a section about the open-source aspects of TomoBEAR. We updated the description of the modules and their dependencies with the corresponding tested versions in Supplementary Text 1. Additionally, we added new Supplementary Text 2, describing the development template for a general-purpose module as well as source code and project folders file structures.

1.4 Binning and reconstruction modules

From the supplementary material it was not clear whether TomoBEAR uses IMOD programs under the hood for the BinStack and Reconstruct modules, or if it has its own implementation of these tasks. I was able to check in the source code (and I praise the authors for sharing it), but please clarify in the modules description what is performed by TomoBEAR itself and what is interfaced from other packages.

Thank you for the comment. Following the recommendation, we updated the description of the modules and provided the corresponding dependencies and their versions tested with TomoBEAR in Supplementary Text 1.

1.5 Deposition of apoferritin dataset

As the authors used two other datasets deposited on EMPIAR for testing TomoBEAR, it would also make sense to deposit in that same repository the purified human apoferritin dataset they generated. This dataset could be useful for the testing and benchmarking of new tools by the community.

Thank you, we deposited both raw data and the resulting structure: EMPIAR-11543 / EMD-17232.

1.6 Cellular tomography data

The authors have tested TomoBEAR on 3 datasets, all of which were either from purified proteins or isolated membranes, containing gold beads for tilt series alignment. While it's clear that TomoBEAR supports tilt series alignment without gold fiducials via patch tracking in IMOD or AreTomo, I would recommend if possible that the authors include one more dataset from in situ cryo-ET, i.e. from

entire cells or FIB-milled lamellae, without gold fiducials. The authors may already have such a dataset in house, or could take one from the EMPIAR. This would strengthen the appeal of TomoBEAR to the in situ cryo-ET community even further, however, it is not essential for the acceptance of the manuscript.

At a minimum, the authors should discuss what differences are expected when processing this kind of data. For example: are processing times significantly different? Can the automated workflow yield good quality reconstructions? How well is the new GPU-enabled template matching expected to work in crowded cellular environments? etc.

Yes, thank you, as a response to this comment and to the reviewer 1 we added a plasma-FIB-milled dataset and performed structural analysis of a 80S ribosome from it. The workflow is overall similar other than tilt series alignment that has to be done without gold beads. We provide a new workflow that can use patch tracking from IMOD and our newly implemented addition of AreTomo. We also added particle picking by crYOLO, however the better structure in this case was obtained by template matching. We wrote a large section about this in the end of the results part in the main text of the manuscript.

To answer the reviewer's question - even though this fiducial-less workflow does not need manual gold beads refinement as in the fiducial-based alignment cases, the workflow still may need some level of manual quality control and parameter optimization. Nevertheless, the newly added Supplementary Figure S1 shows that automatization of fiducial-less workflow in TomoBEAR allows reaching the general processing time on the same scale as for fiducial-based workflows, which could be tested by the users.

2 MINOR COMMENTS

2.1 SUSAN

Instead of referring to unpublished work, the authors could deposit the SUSAN source code, which is already available, in a citable repository such as Zenodo. I look forward to the manuscript describing the SUSAN package.

We are working on releasing the package, to have a snapshot of a functional version we uploaded SUSAN on Zenodo: <https://zenodo.org/record/7950904>. In the current manuscript we do not use the high-resolution refinement with SUSAN and only SUSAN for pre-alignment and the reconstruction of subtomograms by skipping the reconstruction of full tomograms.

2.2 Reorder rows in Table 1

The rows "Number of tomograms", "Number of views per tomogram" and "Manual interventions" would be more useful if listed on the top of the table. Likewise, the hardware settings could be moved to the bottom.

Thank you, we updated it.

2.3 Readability

The manuscript contains some redundant wording and sentences that seem incomplete, as well as a few typos. For better readability, I suggest the authors revise the text to minimize these issues. Below are some examples.

Thank you, we worked on the further improvement of the readability of the manuscript, many modifications are tracked in the updated version of the manuscript.

ABSTRACT

LINES 27-29

(...) to produce high resolution [structures?] with minimal human intervention. TomoBEAR is an open-source and extendable package [that might] accelerate the adoption of in situ structural biology by cryo-ET.

Thank you, we updated it.

INTRODUCTION

LINES 41-42

allowed (...) allowing

Thank you, we updated it.

RESULTS

FIGURE 1 LEGEND

LINE 107

highlighted in red highlight

Thank you, we updated it.

LINE 96

module-specific sections with module-specific parameters

Thank you, we updated it.

LINE 225

tilt-serie to tilt-serie

Thank you, we updated it.

DISCUSSION

LINE 302

Remove "use of" (before "streamlining")

Thank you, we updated it.

--

Reviewed by Ricardo D. Righetto

Reviewers' Comments:

Reviewer #1:

Remarks to the Author:

The authors have satisfied the concerns of this referee. Specifically, the manuscript has been improved by implementing Ctffind4 for CTF determination, IsoNet for deconvolution/filtering and filling of the missing wedge, and crYOLO for neural network-assisted particle picking. Furthermore, by demonstrating the applicability of TomoBEAR to FIB-milled cryo-ET data has expanded the usage of tomoBEAR to fiducial-less datasets which will be useful to the community. The YouTube tutorial videos are an excellent resource to lower the barrier to entry for using tomoBEAR.

Reviewer #2:

Remarks to the Author:

REVIEW NCOMMS-23-01605 – REVISION #2

The revised version of the manuscript by Balyschew, Yushkevich et al is greatly improved compared to the initial submission. They offer important clarifications and additional useful information to potential users and developers of TomoBEAR. Most interestingly, the manuscript now demonstrates the applicability of TomoBEAR for processing cellular cryo-ET data without gold beads (FIB-milled HeLa cells). All my comments and concerns have been properly addressed. The new version could use only a few minor clarifications outlined below. At any rate, I definitely recommend the manuscript for publication.

(All line numbers correspond to the tracked changes version of the main text and supplementary material)

MANUSCRIPT

- LINES 136, 555, Suppl. 129:

Update Dynamo wiki links (L555 and Supp. L129):

https://www.dynamo-em.org//w/index.php?title=Walkthrough_on_GUI_based_tilt_series_alignment

- LINE 100 and elsewhere:

The export of StA projects from TomoBEAR to RELION specifically assumes the v4 version of the latter. This should be made clear from the first mention (and not only later in the text, as is currently) because the behavior of subtomogram averaging in RELION has changed substantially from v3 to v4 (e.g. regarding how CTF is treated). For example, Warp/M, which is widely used in the field, only supports RELION-3.

- LINES 380-381:

This sentence does not add information to the discussion and could be removed as it only refers to unpublished data:

Steps of TomoBEAR scale linearly with the data and can be used for larger data sets, over 500 tomograms (Kudryashev group, unpublished).

- LINE 413:

TomoBEAR has been demonstrated to work well for data not containing gold fiducials, so it seems unnecessary to say it is particularly suited for samples containing them.

- LINE 592:

What structure was used as a template here?

- Fig S1 legend wording:
"tomograms post-processing of tomograms"

- Supp. Text 3 title:
EMPIAR-10064 (remove the extra "0")

REBUTTAL LETTER

- I appreciate that the authors have followed my advice to implement a denoising module in TomoBEAR, supporting IsoNet and the NAD filter which are widely applicable. However, there seems to be a misunderstanding about the input data required by cryo-CARE that I would like to clarify. It is not necessary to align the tilt series from odd/even frames separately. Briefly, this is how a tilt series can be prepared for cryo-CARE denoising:

1. Motion correction: align the raw movie frames, as usual
 2. Average the aligned frames per tilt and assemble a tilt series, as usual
 3. In addition: average separately only the odd and the even frames per tilt (already motion corrected above), and assemble the respective odd/even tilt series
 4. Perform alignment of the conventional, full-frame tilt series from step #2 (using fiducials if applicable, or patch tracking in IMOD, or AreTomo, etc)
 5. Apply the alignment parameters obtained in step #4 to the odd/even TS, thus enabling the reconstruction of odd/even tomogram pairs that are identical except for the noise.
- For an example of how this step can be implemented, please see the cryocare_pipeline script at: <https://github.com/CellArchLab/cryoet-scripts>

It is true that the logistics required are relatively complicated, but a pipeline manager such as TomoBEAR is expected to facilitate precisely cases like this. I am confident that cryo-CARE can be incorporated in a future version of the package, and its current absence does not impact the publication of the work in any way.

--

Reviewed by Ricardo D. Righetto

Reviewer #1 (Remarks to the Author):

The authors have satisfied the concerns of this referee. Specifically, the manuscript has been improved by implementing Ctffind4 for CTF determination, IsoNet for deconvolution/filtering and filling of the missing wedge, and crYOLO for neural network-assisted particle picking. Furthermore, by demonstrating the applicability of TomoBEAR to FIB-milled cryo-ET data has expanded the usage of tomoBEAR to fiducial-less datasets which will be useful to the community. The YouTube tutorial videos are an excellent resource to lower the barrier to entry for using tomoBEAR.

We thank the reviewer for the useful constructive review.

Reviewer #2 (Remarks to the Author):

REVIEW NCOMMS-23-01605 – REVISION #2

The revised version of the manuscript by Balyschew, Yushkevich et al is greatly improved compared to the initial submission. They offer important clarifications and additional useful information to potential users and developers of TomoBEAR. Most interestingly, the manuscript now demonstrates the applicability of TomoBEAR for processing cellular cryo-ET data without gold beads (FIB-milled HeLa cells). All my comments and concerns have been properly addressed. The new version could use only a few minor clarifications outlined below. At any rate, I definitely recommend the manuscript for publication.

We thank the reviewer for the useful and constructive suggestions. We followed

(All line numbers correspond to the tracked changes version of the main text and supplementary material)

MANUSCRIPT

• LINES 136, 555, Suppl. 129:

Update Dynamo wiki links (L555 and Supp. L129):

[https://www.dynamo-em.org//w/index.php?title=Walkthrough on GUI based tilt series alignment](https://www.dynamo-em.org//w/index.php?title=Walkthrough%20on%20GUI%20based%20tilt%20series%20alignment)

Thank you, as this part is not released and we did not have a DOI for it – we replaced the reference to the wiki by a reference to an original article.

• LINE 100 and elsewhere:

The export of StA projects from TomoBEAR to RELION specifically assumes the v4

version of the latter. This should be made clear from the first mention (and not only later in the text, as is currently) because the behavior of subtomogram averaging in RELION has changed substantially from v3 to v4 (e.g. regarding how CTF is treated). For example, Warp/M, which is widely used in the field, only supports RELION-3.

Thank you for an insightful comment, we changed RELION to RELION4 through the text.

- **LINES 380-381:**

This sentence does not add information to the discussion and could be removed as it only refers to unpublished data:

Steps of TomoBEAR scale linearly with the data and can be used for larger data sets, over 500 tomograms (Kudryashev group, unpublished).

We removed the sentence.

- **LINE 413:**

TomoBEAR has been demonstrated to work well for data not containing gold fiducials, so it seem unnecessary to say it is particularly suited for samples containing them.

Thank, we cut down the specific mention of the gold fiducials.

- **LINE 592:**

What structure was used as a template here?

Thank you, we specified the used template EMD-15636.

- **Fig S1 legend wording:**

“tomograms post-processing of tomograms”

Thank you, we updated it.

- **Supp. Text 3 title:**

EMPIAR-10064 (remove the extra “0”)

Thank you, we updated it.